# Energy Management System Optimization of Drug Store Electric Vehicles Charging Station Operation

Yongyi Huang [1,*], Atsushi Yona [1], Hiroshi Takahashi [2], Ashraf Mohamed Hemeida [3], Paras Mandal [4], Alexey Mikhaylov [5], Tomonobu Senjyu [1] and Mohammed Elsayed Lotfy [1,6,*]

1 Department of Electrical and Electronics Engineering, University of the Ryukyus, Okinawa 903-0213, Japan; yona@tec.u-ryukyu.ac.jp (A.Y.); b985542@tec.u-ryukyu.ac.jp (T.S.)
2 Fuji Electric Co., Ltd., Tokyo 191-0064, Japan; takahashi-hirosi@fujielectric.com
3 Electrical Engineering Department, Faculty of Energy Engineering, Aswan University, Aswan 81528, Egypt; ashraf@aswu.edu.eg
4 Department of Electrical and Computer Engineering, University of Texas at El Paso, El Paso, TX 79968, USA; pmandal@utep.edu
5 The Russian Federation, Financial University, Moscow 124167, Russia Federation; ayumihajlov@fa.ru
6 Electrical Power and Machines Department, Zagazig University, Zagazig 44519, Egypt
* Correspondence: w120810111@gmail.com (Y.H.); Mohamedabozed@zu.edu.eg (M.E.L.)

**Abstract:** Electric vehicle charging station have become an urgent need in many communities around the world, due to the increase of using electric vehicles over conventional vehicles. In addition, establishment of charging stations, and the grid impact of household photovoltaic power generation would reduce the feed-in tariff. These two factors are considered to propose setting up charging stations at convenience stores, which would enable the electric energy to be shared between locations. Charging stations could collect excess photovoltaic energy from homes and market it to electric vehicles. This article examines vehicle travel time, basic household energy demand, and the electricity consumption status of Okinawa city as a whole to model the operation of an electric vehicle charging station for a year. The entire program is optimized using MATLAB mixed integer linear programming (MILP) toolbox. The findings demonstrate that a profit could be achieved under the principle of ensuring the charging station's stable service. Household photovoltaic power generation and electric vehicles are highly dependent on energy sharing between regions. The convenience store charging station service strategy suggested gives a solution to the future issues.

**Keywords:** V2H; MILP; energy management; optimization; energy economics

## 1. Introduction

With the changing trend of the global climate, people have also realized the importance of environmental protection [1,2]. A report from the highly credible Intergovernmental Panel on Climate Change (IPCC) pointed out what are the likely consequences of the 1.5 °C increase of the Earth's temperature [3]. The ensuing consequences are various occurrences of extreme weather, rising sea-levels, species extinction, etc. In order to reduce carbon emissions, the development and utilization of clean energy have been rapidly developed today [4,5]. In addition, in the transportation field, carbon emissions and air pollution's health effects on the human body caused by internal combustion engines [6] cannot be ignored. Electric vehicles began in the 19th century, and, due to the continuous development of technology, the number will continue to increase in this century [7]. However, due to imperfect infrastructures, it has not yet become popular.

The work in Reference [8] mentioned energy storage and electric vehicles (EVs) will make up for the intermittency of wind and solar power generation, which is essential for decarbonization and the improvement of a renewable energy utilization ratio. About the fast charging problem, a number of researchers have also invested in research [9,10]. Bunker et al. [11] takes a look at a number of small, islanded microgrids from around the

world, sharing examples of communities moving from a single resource to a complex range of resources, such as wind, solar, biodiesel, hydro, and energy storage. Energy storage is mentioned as an important component of renewable-energy-based islanded and remote community microgrids. In the form of a grid-connected microgrid, Akram et al. [12] proposed a technique for mutual capacity management of hybrid renewable power generation systems and energy storage. The integrated optimization takes advantage of both the advantages of hybrid power generation and the advantages of hybrid energy storage systems (HESS). The other side will be the charging demand of EVs. In this regard, the future is not only the grid supplying the EVs (G2V), but also the EVs supplying power to the grid (V2G) or houses (V2H) [13–16]. Due to the promotion of renewable energy, the future energy storage industry will definitely need to strengthen its development. In this process, electric vehicles can be regarded as mobile power sources. Hence, the development of EVCS has become an urgent need.

Su et al. [17] considers the battery degradation cost, charging cost, and waiting time from the user's view, and uses artificial intelligence fish swarm algorithm to establish an optimization model for car charging. This significantly reduces the cost of battery degradation and the cost of electricity without negatively affecting the grid- considering that most private cars will be charged at night at home. Reference [18] mainly focuses on the impact of Plug-in Electric Vehicle (PEV) charging on residents' electricity consumption. Using a high-resolution (10-min resolution) residential electricity demand and PEV usage model, this method can quantify the customer's energy usage behavior and the actual world vehicle usage rate. This ensures the safe and stable operation of the power grid. Regarding the construction of charging stations, Mehrjerdi et al. [19] designed a random model of EVCS integrated with wind energy. Considering the uncertainty of wind energy, the optimization method of mixed-integer linear programming is used. The optimal combination of charging facilities and the size and operation of the storage system of the electric vehicle charging station (EVCS) is obtained. Luo et al. [20] considered that charging facilities with different rated charging powers can meet the changing needs of different EV owners, while also taking into account the temporal and spatial distribution of EV charging demand and other factors. Moreover, Knirsch et al. [21] wrote a privacy-preserving blockchain-based electric vehicle charging method with dynamic tariff decisions. It provides a way for various power suppliers and charging station operators to provide energy at variable prices based on supply and demand.

On the other hand, in the development and utilization of solar energy, Japan's Feed-in Tariff (FiT) system has increased the proportion of renewable energy significantly. Solar energy installation and usage, as well as smart homes that are connected to the grid, are among them [22,23]. The amount of electricity produced has increased significantly as a result of the installation of a large number of solar power generation systems. This could result in the following 3 outcomes, some of which are noticeable at work [24–26].

- First, due to the unpredictability of environmental conditions, power generation can spike up or down over specific time intervals, causing the frequency and distribution voltage of the original power system to rapidly fluctuate.
- Second, the rapid deployment of solar power generation systems has resulted in an excessive increase in power supply during the day, resulting in the duck curve issue.
- Third, the interconnection between photovoltaic power generation systems and power systems will hit a breaking point in some places.

Some research has looked into the impact of the FiT policy on PV systems [27,28]. As a result, the cost of photovoltaic power generation is expected to decline, potentially reducing government payments to producers.

We now encounter two problems: Firstly, the rapid popularity of electric vehicles will require more charging stations to be put into operation. Secondly, household photovoltaic power generation needs to be consumed nearby. Coincidentally, it is worth noting that drug stores (corner stores/convenience stores) are common. Sanders et al. [29] suggests that new policies are needed to promote a healthier retail environment for youth, due

to approximately 4.1 million America adolescents visiting convenience stores weekly. This shows the great influence of convenience stores. Moreover, their locations are usually located in a high traffic areas, the goods are enticing, and more importantly, there are usually many houses around. They also have a role to provide supplies in emergency situations. Especially, Okinawa often suffers from typhoons in the summer. They sometimes cause a variety of inconveniences, such as power outages, communication disruptions, and road blockages. At this time, the drug stores near residences help to supply food, water, and electricity to the surrounding residents.

This study considered a Community Energy Management System (CEMS) to solve those problems. The drug store collects unused solar power from surrounding households at an appropriate price, resells it to EVs, then sells the remaining power to other households. Taken into consideration is the uncertainty of the weather, household electricity consumption, EVs charging time, and so on. This study uses weather data throughout the year in Okinawa, considers the householder power consumption errors at different times, and charging peak of electric vehicles, etc., and then various scenarios are simulated. The impact of various uncertainties are shown as much as possible. Mixed Integer Linear Programming (MILP) is used to show the demonstration of the feasibility of this plan. The schema is shown in Figure 1. The main contributions of this article include the following 3 points:

- Added a suggestion for selling goods in drug stores (corner stores or convenience stores) and demonstrated the feasibility: selling renewable electricity energy.
- This article proposes a plan for household photovoltaic power generation to be used and sold nearby. It solves the current problem of low photovoltaic power generation on the roof of homes and low electricity recycling prices. It improves the homeowner's income while improving energy utilization.
- Based on the convenience of the geographical location of drug stores, a plan to establish a charging stations for electric vehicles using renewable energy is proposed. This provides an energy supply service guarantee for the coming electric vehicle era.

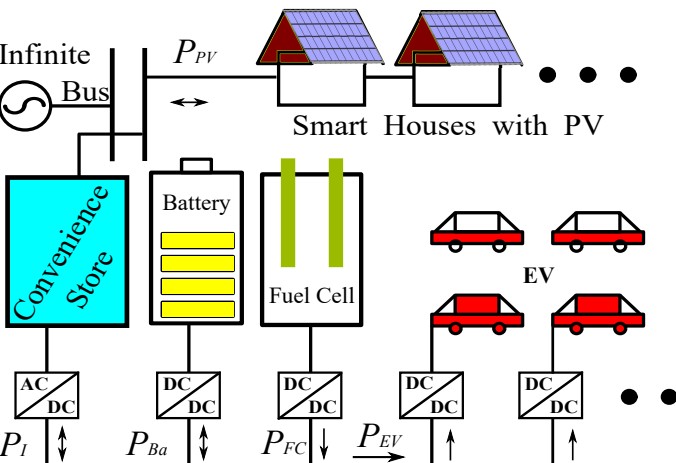

**Figure 1.** Community energy management model.

The following explains the arrangement of this article. In Section 2, analyze the components and restrictions of this model, and use mathematical formulas to express them. The naming list is given below to help understand this section. Section 3, first, estimates the usage status of charging stations based on the characteristics of people's habits. Then, taking into account the user's electricity consumption and power generation, estimate the amount of electricity that can be provided. Finally, based on the electricity consumption and solar radiation of Okinawa in 2019, the electricity price for selling electricity to trams and purchasing photovoltaics was specified. In Section 4, the results of this simulation are shown. The summary and outlook are in Section 5.

## 2. Community Energy Management System Model

When considering the layout requirements of the charging station, it mainly meets two aspects, demand and possibility. The degree of demand for charging stations is mainly reflected in traffic flow and convenience. The possibilities are mainly reflected in environmental protection and regional power distribution capabilities. This section mainly elaborates on feasibility and is also the focus of this article.

The convenience store is connected to smart houses which receives or regulates the output power by the PV shown in Figure 1. There are batteries to store unused electric energy, and a fuel cell which is designed to provide sufficient power to EVs when the solar generation is insufficient. Here, assume that a smart house family within 5000 m of the convenience store will sign an agreement with the convenience store. The charging station is designed to simultaneously supply 5 EVs with 50 kW of fast charge supply.

### 2.1. Objective Function

The purpose of the plan is to maximize the benefits of charging stations based on the impact of weather and various uncertain factors. Simply put, it is the daily income minus the daily investment cost. Therefore, the following objective function can be derived Equation (1).

$$max\ C_{day} = \sum_{i=1}^{24} \{C_{EV}(i)P_{EV}(i) + C_{st}(i)P_{st}(i) - C_{Pt}(i)P_{Pt}(i) - C_{FC}(i)P_{FC}(i) - D_{Ba} - D_{FC}\}. \tag{1}$$

### 2.2. Restrictions

In this CEMS model, shown as Figure 1, the $P_I$ is the current of the connection point, the $P_{Ba}$ is the charge or discharge power of the battery, and the $P_{FC}$ is the fuel cell output power. The $P_{EV}$ is charging consumption of EV. Table 1 displays the parameters of battery and fuel cell. The relationship between them satisfies the following formula:

$$P_{EV}(i) = P_I(i) + P_{Ba}(i) + P_{FC}(i). \tag{2}$$

Restrictions on power flow changes at connection points:

$$|P_I(i) - P_I(i-1)| < P_{Imax}. \tag{3}$$

Battery charge or discharge capacity limit:

$$|P_{Ba}(i)| \leq P_{Bamax}. \tag{4}$$

State of charge limit:

$$20\% < \epsilon < 80\%. \tag{5}$$

Fuel cell charge capacity limit:

$$|P_{FC}(i)| \leq P_{FCmax}. \tag{6}$$

**Table 1.** Battery and fuel cell parameters.

| Item | Battery | Fuel Cell |
|---|---|---|
| Capacity [kWh] | 20 | 20 |
| Number | 15 | 1 |
| Rated Charging Power [kWh] | 10 | 20 |
| Charging Efficiency | 90% | 40% |
| Price [JPY] | $1.2 \times 10^7$ | $2 \times 10^8$ |
| Lifetime [Year] | 15 | 20 |

### 3. Simulation Conditions Scenarios

Since Okinawa's weather is unpredictable, and the number of people traveling during the holidays is lower, power consumption and EV load would fluctuate. This segment will go into different situations for energy demand and generation, as well as the specifics of how electricity prices are calculated. Enumerating all the possible scenarios throughout the year.

#### 3.1. EV Charging Station

The operating modes of EVs are mainly divided into buses, taxis, family cars, and official vehicles. Buses have prescribed routes and unified stations. They are more suitable for battery replacement during operation, and the replaced batteries are charged at night. For taxis, due to the high frequency of use during the day, the tight charging time, and the distance uncertainty, you can choose to fast charge or replace the battery. For official vehicles, the parking time during the day is short and the frequency of commuting is high. Private vehicles are parked longer than official vehicles, making them more suitable for slow charging. Transportation vehicles are used a lot during the day and are driven long distances.

Zhang et al.'s [30] stochastic models of taxis, buses, charging stations, and battery exchange systems are established. Subsequently, the service capabilities of EVSE will be compared. The influence of factors, such as the size of the battery, the speed of the vehicle, the power of the charging station, and the price of the exchange service on the service capacity, are studied. Santos et al. [31] reports the Distribution of Vehicle Trips by Trip Purpose and Start Time of Trip. Regardless of the purpose of use, the traffic distribution of all vehicles is shown in Figure 2 as the red line. Peak travel is between 17:00 and 18:00, the second time period is noon, and the third is between 06:00 to 07:00.

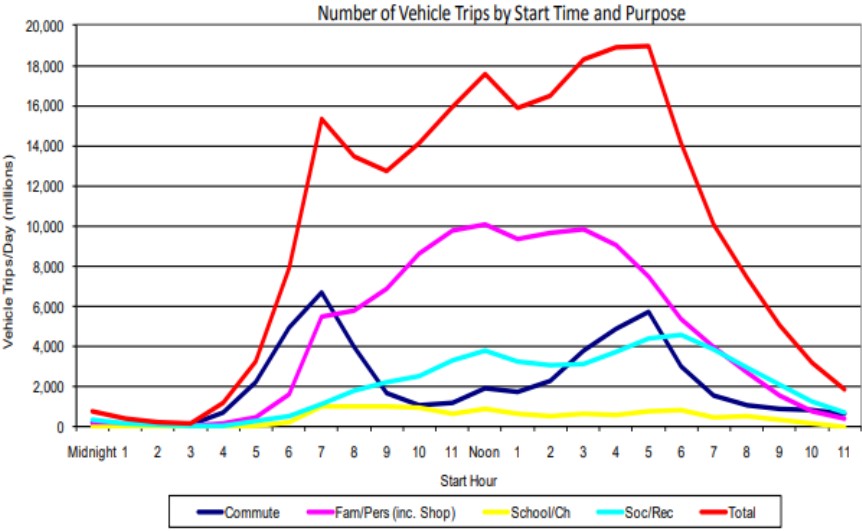

**Figure 2.** Distribution of vehicle trips by trip purpose and start time of trip [31].

Hence, it is anticipated that all 5 charging ports of a charging station have a high probability of being used during the peak travel period. During a normal travel period, one less charging port is more likely to be used. Here, the main peak of charging is around 12:00, and the second peak is in the return journey, getting off work, and it is around 07:00 to 09:00 and 15:00 to 17:00. In the evening, off-peak hours are set from 22:00 to 04:00. In addition, suppose that, during the holidays, the usage of charging stations is half of the normal situation. The 365 charging scenarios of EV are shown in Figure 3.

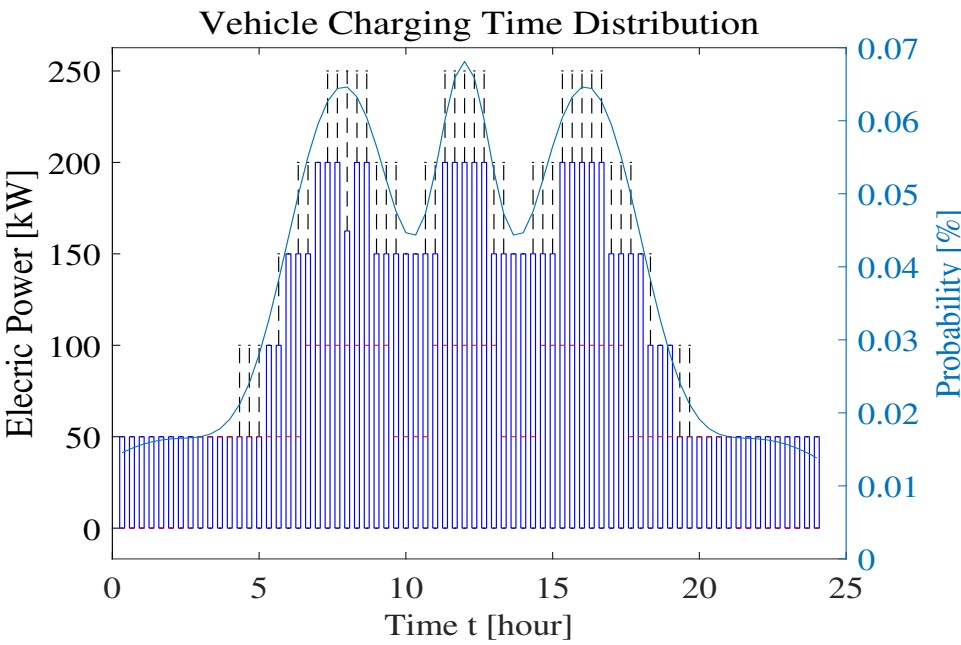

**Figure 3.** EV charging 365 scenarios with holiday.

### 3.2. Household Consumption

When a household produces more electricity than needed, the remaining electricity will be sold to a convenience store to charge the EVs. Under the premise of fully considering uncertainties, the electricity consumption as shown in Figure 4 is obtained. The basis of the design is shown in the Appendix A, Table A1 [32]. The red number is the consumption during holidays. From 00:00 to 08:00, we find a fluctuation range of 50 W in energy consumption per hour, and, in other periods, it is 100 W.

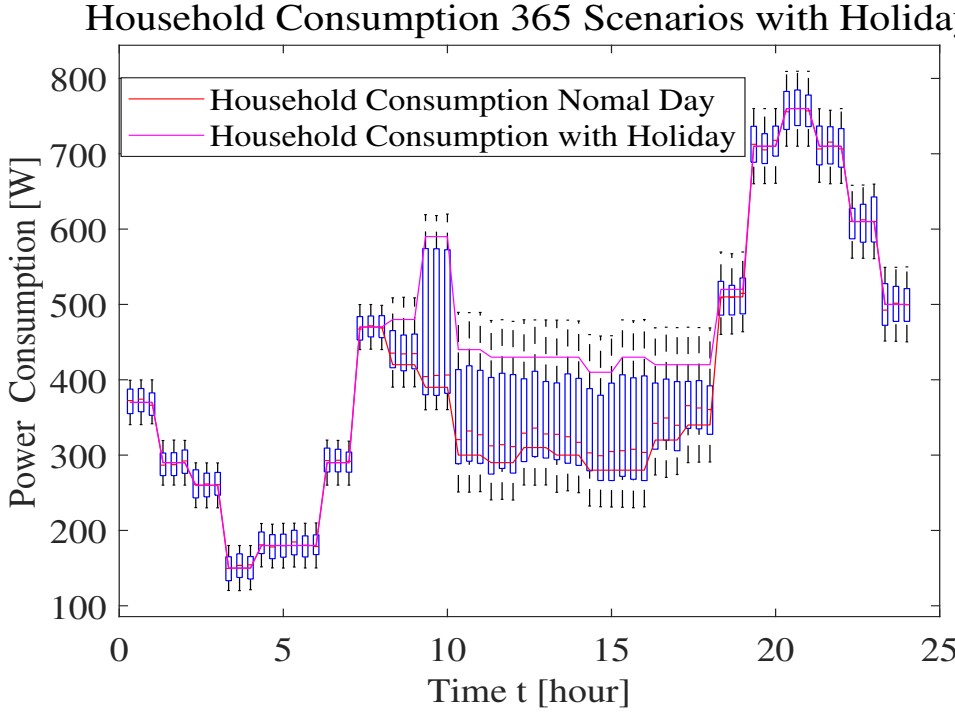

**Figure 4.** One-year scenarios with holiday.

### 3.3. Real-Time Price of Electricity Demand

Due to the difference in the peak power consumption of each household and the impact of the holiday on the power consumption, here, the charging price of EVs is determined based on the electricity consumption in Okinawa that day [33]. The price set is shown in Equation (7). The $l$ [MW] is the entire consumption of the Okinawa at that time. Figure 5 is box-plot of 365 scenarios' hourly selling price.

$$C_{EV} = \begin{cases} 10, & 0 \le l \le 500 \\ 0.2(l - 500) + 10, & 500 \le l \le 1500. \\ 30, & 1500 \le l \end{cases} \tag{7}$$

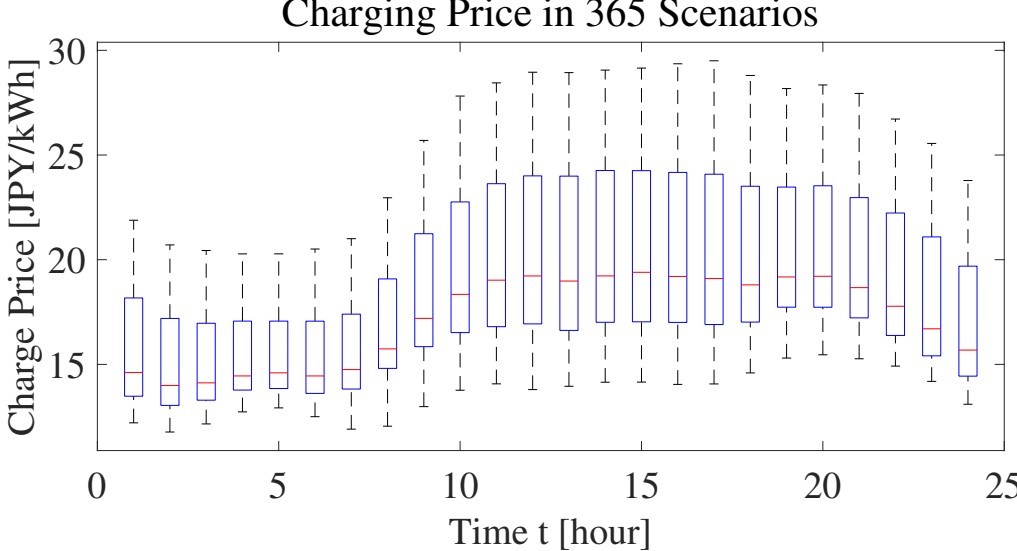

**Figure 5.** Electricity selling price.

In addition, the changes of solar radiation will give some impact on PV power generation. When there is strong light from the sun, the convenient charging station will receive a large amount of electricity. During those times, to protect the charging station and lower the cost, the purchase price of electricity will be reduced. On the contrary, when the light is insufficient, the purchase price of electricity will be increased. From Equation (8), the purchase price of electricity can be obtained as shown in Figure 6.

$$C_{Pt} = \frac{5}{\exp\left(\frac{P_{Pt}}{100}\right)} + 10. \tag{8}$$

Figure 7 shows the flow chart of this scheme. First, input PV power generation and residential electricity consumption, and then calculate the trade-able power. The energy is managed through the drug store, and the excess energy is stored in the battery. When the energy is insufficient, it is provided by the fuel cell. Then, input various prices. The whole process is optimized by MATLAB's MILP toolbox. Finally, loop for one year and output the result.

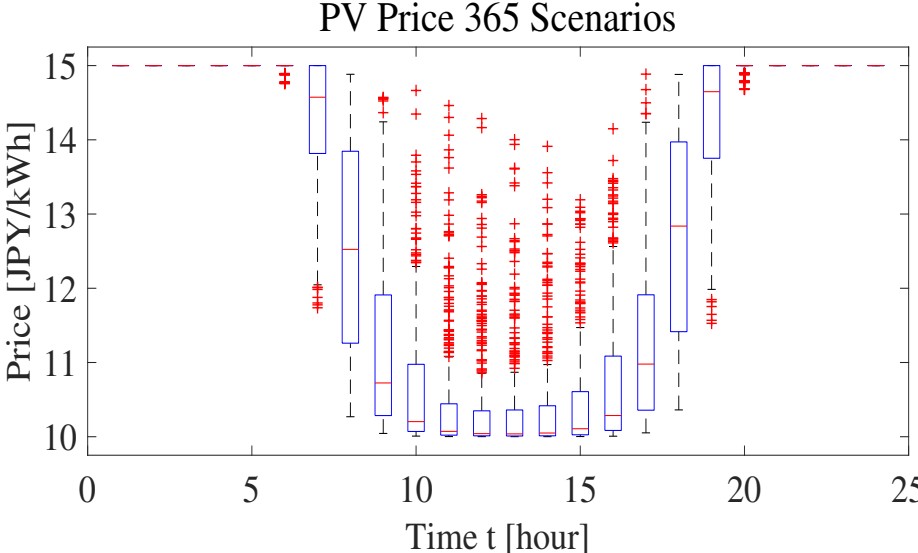

**Figure 6.** Purchase PV power price.

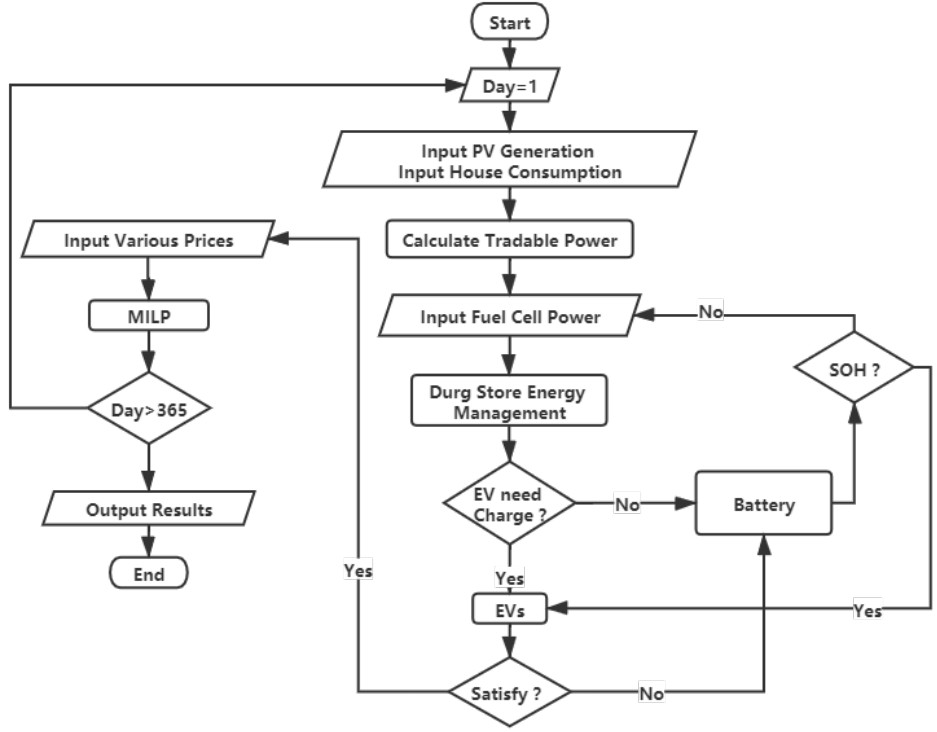

**Figure 7.** Flow chart of scheme.

## 4. Simulation Results Discussion

After optimization, the results of one year's operation of the charging station were obtained. Figure 8 showed the profit or loss of all scenarios in the whole year.

It can be seen that the most profitable day is the 197th day, 124,152 JPY, and the least profitable day is the 264th day, 1615 JPY.

Figure 9 displays the details of two days. The left column is the most profitable day (197th), and the right is the least profitable day (264th). Figure 9a,b is the charging situation and electricity price of that day. It can be found that the electricity price on the 197th day is higher than 264th day, and the charging capacity is also significantly higher. Figure 9c,d is the charging and discharging state of the battery. When the demand for charging is high, the frequency of battery utilization is higher. Contrasting the battery state of charge is

shown in Figure 9e,f, where positive means charging for EVs, while minus means accepting external power. Figure 9g,h is the working state of the fuel cell battery. When PV power generation is sufficient, the fuel cell only provides 10kw constant power for convenience stores. Correspondingly, when the PV power generation is insufficient, the fuel cell will work more frequently. Figure 9i,j is the power flow and PV output power. Combined with Figure 9e,f, it can be seen that both the battery and the charging station are working within a safe range.

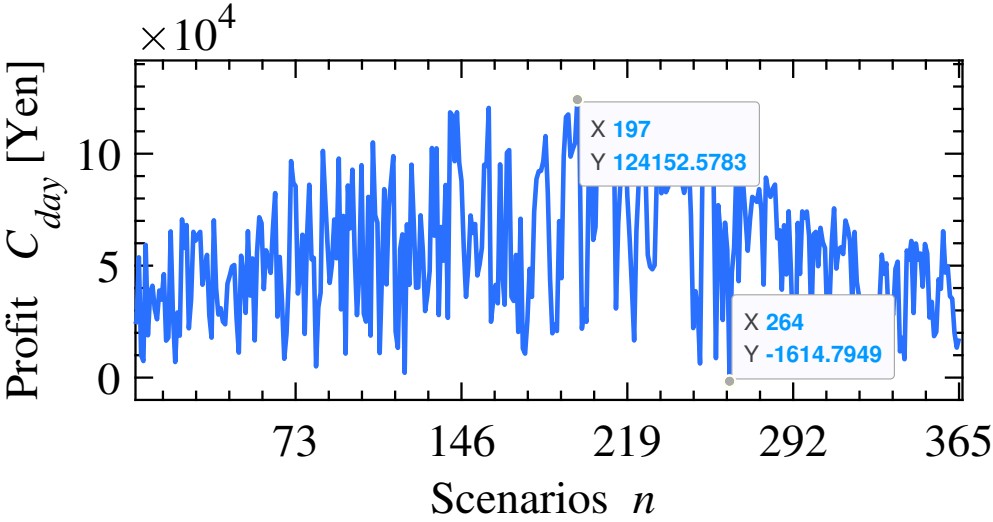

**Figure 8.** One-year profitability.

Due to the arrival of the typhoon on the September 21st (264th day), the sunlight was very weak, the number of travelers were small, and the overall power consumption was low, which made the charging station's interest negative.

In the days when the analysis income is less than 10,000 JPY, as shown in Table 2, it shows that these days also fall on weekends or holidays and rainy days. Figure 10 illustrates the amount of charging situation, charging price, and PV performance these days. It has been discovered that, when the PV output is lacking, the number of travelers is limited, the price of electricity gets low, and the profit will be less than 10,000 JPY. From the electricity selling price, it can be seen that Okinawa's electricity consumption on that day is relatively small.

**Table 2.** Profit less than 10,000 JPY per day.

| Days | 5 | 6 | 20 | 68 | 82 | 121 | 251 | 258 | 293 | 341 |
|---|---|---|---|---|---|---|---|---|---|---|
| Profit [JPY] | 9963 | 7392 | 6956 | 8384 | 4990 | 2135 | 6301 | 8686 | 5301 | 8211 |
| Week | Saturday | Sunday | Sunday | Saturday | Saturday | Thursday | Sunday | Sunday | Sunday | Saturday |
| Date | 5 January | 6 January | 20 January | 9 March | 23 March | 1 May | 8 September | 15 September | 20 October | 7 December |
| Weather | Rainy | Rainy | Rainy | Rainy | Rainy | Rainy | Rainy | Rainy | Rainy | Rainy |

In addition, it can be seen from Figure 8 that the best returns are distributed in summer. Compared with spring and winter, the distribution of lines is relatively sparse and the volatility of returns is relatively small. The reason for the fluctuation of profits can be considered to be caused by the weather and the number of passengers.

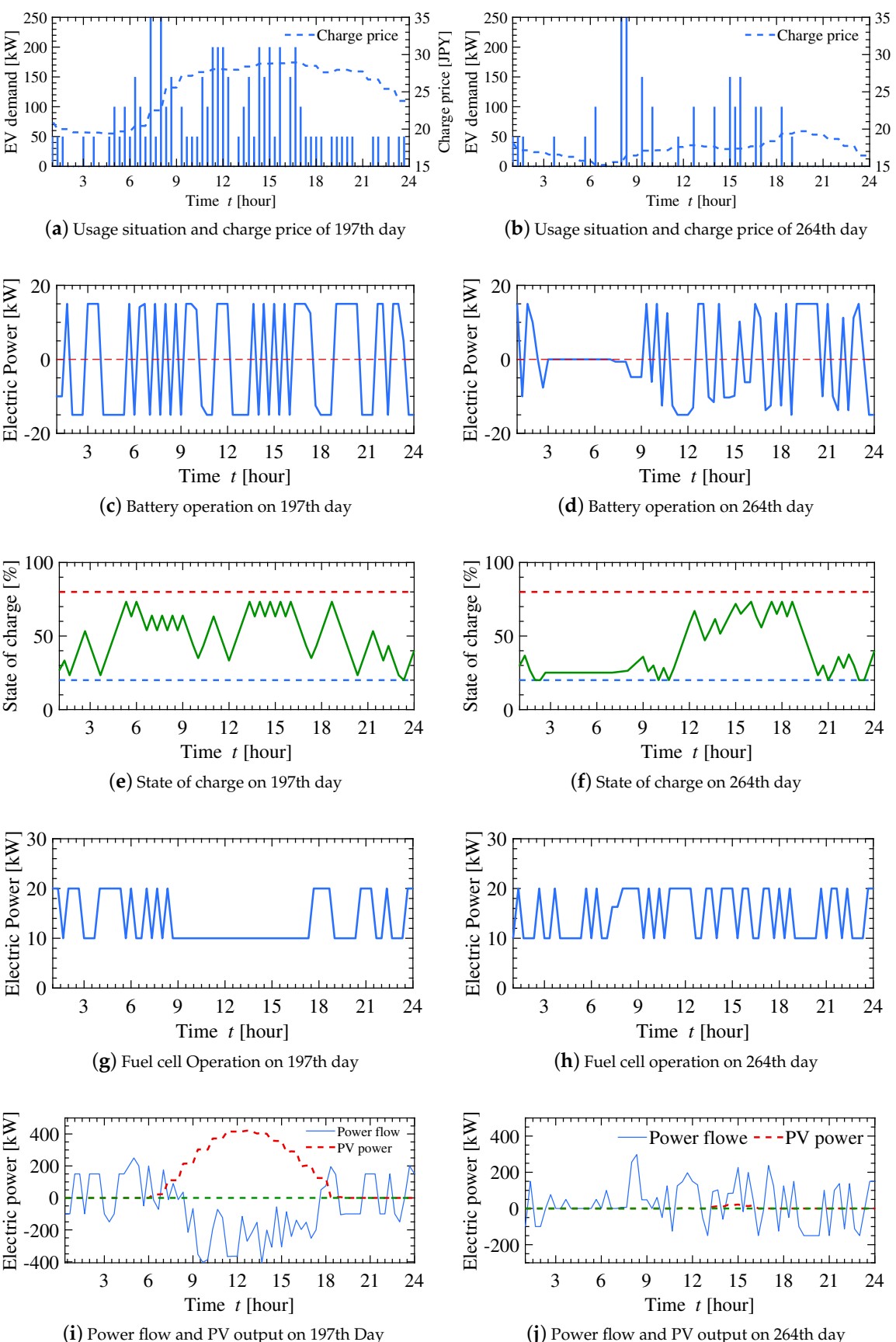

**Figure 9.** The details of day 197 and day 264.

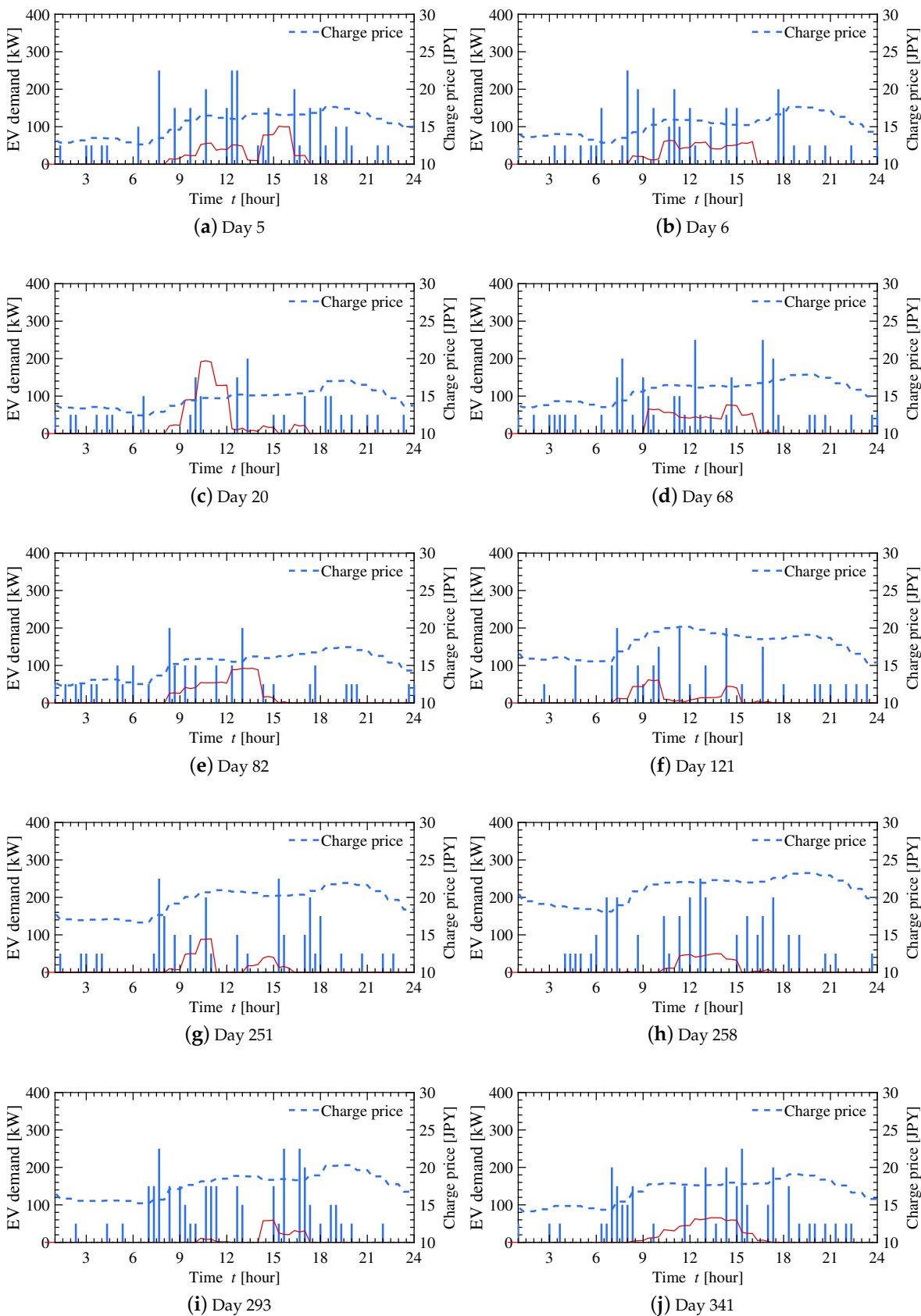

**Figure 10.** Profit less than 10,000 JPY days.

## 5. Conclusions and Outlook

This study investigates and helps solve the issues that have arisen as a result of the widespread use of rooftop photovoltaic power generation. When the FiT scheme is phased out, the FiT price for residential solar power generation will be reduced. At the same time, the popularity of EVs is on an unstoppable rise, which is being accompanied by a slew of issues, such as energy supply, charging station design, and location range. According to the article's solution, convenience stores sign an energy transaction contract with surrounding residents. The residents sell the surplus electricity convenience stores. The convenience store collects the electricity and sells it to the EV users at a reasonable cost. With a complete understanding of the different risks involved and with adequate data support, this paper increases the profitability of retail stores and raises residents' income, thus resolving the energy supply and management problem of the microgrid. The various scene-settings show the feasibility and efficacy of this plan under a variety of unpredictable circumstances.

Panchal et al. [34] introduced the research progress of dynamic and static wireless charging, and after comparing the difference between plug-in charging and wireless charging, he demonstrated the advantages of wireless charging. In terms of convenience, the popularity of electric vehicles will rapidly increase in the future. But, also due to convenience, it may cause people to charge irregularly. Therefore, a related charging system was formulated to cut peaks and fill valleys. This will greatly reduce the peak-to-valley gap of the power grid. In addition, an electric car can be regarded as a battery when not in use. The battery can effectively alleviate the uncertainty of renewable energy. A reasonable combination of electric vehicles and renewable energy may be the solution to the duck curve problem [35–38].

**Author Contributions:** Conceptualization, Y.H. and M.E.L.; methodology, Y.H.; software, Y.H.; validation, M.E.L. and A.Y.; formal analysis, H.T.; investigation, A.M.H.; resources, P.M.; data curation, A.M.; writing–original draft preparation, Y.H. and A.M.; writing–review and editing, M.E.L., A.Y., H.T., A.M.H., P.M., A.M., and T.S.; visualization, A.M.; supervision, T.S.; project administration, M.E.L. and T.S.; funding acquisition, T.S. All authors have read and agreed to the published version of the manuscript.

**Funding:** This research received no external funding.

**Institutional Review Board Statement:** Not applicable.

**Informed Consent Statement:** Not applicable.

**Data Availability Statement:** Not applicable.

**Acknowledgments:** The authors gracefully acknowledge Harun Or Rashid Howlader, Hasan Masrur, and John for the insightful discussions throughout the course of this article.

**Conflicts of Interest:** The authors declare no conflict of interest.

## Abbreviations

The following abbreviations are used in this manuscript:

| | |
|---|---|
| MILP | Mised Integer Liner Programming |
| IPCC | Intergovernmental Panel on Climate Change |
| EVs | Electric Vehicles |
| HESS | Hybrid Energy Storage Systems |
| FiT | Feed-in Tariff |
| CEMS | Community Energy Management System |
| JPY | Japanese Yen |

## Appendix A

Approximate power consumption per day for an average family of 2 people.

**Table A1.** Daily power consumption.

| Time [h] | Toilet [W] | Standby [W] | Lighting [W] | TV [W] | Fridge [W] | AC [W] | Other [W] | Total [W] |
|---|---|---|---|---|---|---|---|---|
| 0 | 10 | 30 | 60 | 10 | 20 | 200 | 40 | 370 |
| 1 | 10 | 30 | 40 | 10 | 20 | 150 | 30 | 290 |
| 2 | 0 | 40 | 30 | 0 | 20 | 150 | 20 | 260 |
| 3 | 0 | 30 | 20 | 0 | 20 | 50 | 30 | 150 |
| 4 | 10 | 40 | 30 | 0 | 20 | 50 | 30 | 180 |
| 5 | 10 | 30 | 20 | 10 | 20 | 50 | 40 | 180 |
| 6 | 10 | 40 | 30 | 30 | 20 | 100 | 60 | 290 |
| 7 | 10 | 30 | 40 | 60 | 20 | 200 | 110 | 470 |
| 8 | 10 | 30 | 40 | 60/100 | 20/40 | 100 | 160 | 420/480 |
| 9 | 10 | 30 | 40 | 70/100 | 20/40 | 50/100 | 170 | 390/590 |
| 10 | 10 | 30 | 40 | 60/100 | 20 | 0/100 | 140 | 300/440 |
| 11 | 10 | 30 | 40 | 60/100 | 30 | 0/100 | 120 | 290/430 |
| 12 | 10 | 30 | 70 | 80/100 | 30 | 0/100 | 90 | 310/430 |
| 13 | 10 | 30 | 70 | 70/100 | 20 | 0/100 | 100 | 300/430 |
| 14 | 10 | 30 | 70 | 70/100 | 20 | 0/100 | 80 | 280/410 |
| 15 | 10 | 40 | 70 | 70/100 | 20/40 | 0/100 | 70 | 280/430 |
| 16 | 10 | 30 | 70 | 70/100 | 20/40 | 50/100 | 70 | 320/420 |
| 17 | 10 | 30 | 80 | 70/100 | 20 | 50/100 | 80 | 340/420 |
| 18 | 10 | 30 | 100 | 90/100 | 30 | 150 | 100 | 510/520 |
| 19 | 10 | 40 | 180 | 110 | 30 | 200 | 140 | 710 |
| 20 | 10 | 40 | 180 | 130 | 20 | 250 | 130 | 760 |
| 21 | 10 | 40 | 180 | 120 | 20 | 230 | 110 | 710 |
| 22 | 10 | 40 | 150 | 90 | 20 | 200 | 100 | 610 |
| 23 | 10 | 40 | 120 | 40 | 20 | 200 | 70 | 500 |

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
