# Peer review of "Energy Management System Optimization of Drug Store Electric Vehicles Charging Station Operation"

_sustainability, doi:10.3390/su13116163_

Round 1

Reviewer 1 Report

Work with data is interesting but authors need to improve results discussion and the cooperation with similar works (only 26 papers in a very hot topic shows that authors did not cover properly the state of art)

Authors should also based on the state of art raise a proper research question and discuss based on results achieved

Reviewer 2 Report

This is the review of the manuscript entitled „Energy Management System Optimization of Drug Store Operate Electric Vehicles Charging Station”.

The subject may attract interest to the readers. However, the following requests/suggestions could be taken into account to improve the quality of the manuscript:

  • The novelty of the work must be clearly addressed and discussed, compare your research with existing research findings and highlight novelty.
  • The main objective of the work must be written on the more clear and more concise way at the end of introduction section.
  • Introduction section must be written on more quality way, i.e. more up-to-date references addressed. Research gap should be delivered on more clear way with directed necessity for the conducted research work.
  • The authors can highlight the usefulness of the study in the practical applicability.
  • The provided methodology is quite simple and the proposed configuration is not novel? Please elaborate more detail reasoning for the selected Energy Management Model and provide a more in-depth discussion of results.
  • RESULTS - providing information regarding how the accuracy of the results was verified.
  • The conclusions need to be written better. The authors should highlight 3-5 bullet points that represent the main findings of this work, not general considerations.
  • Conclusion section is missing some perspective related to the future research work, quantify main research findings.

Round 2

Reviewer 2 Report

All my concerns have been addressed.